# High-speed motility originates from cooperatively pushing and pulling flagella bundles in bilophotrichous bacteria

Klaas Bente[1†], Sarah Mohammadinejad[2,3,4†], Mohammad Avalin Charsooghi[1,5], Felix Bachmann[1], Agnese Codutti[1,2], Christopher T Lefèvre[6], Stefan Klumpp[4*], Damien Faivre[1,6*]

[1]Department of Biomaterials, Max Planck Institute of Colloids and Interfaces, Potsdam, Germany; [2]Department of Theory and Bio-Systems, Max Planck Institute of Colloids and Interfaces, Potsdam, Germany; [3]Department of Biological Sciences, Institute for Advanced Studies in Basic Sciences, Zanjan, Islamic Republic of Iran; [4]Institute for the Dynamics of Complex Systems, University of Göttingen, Göttingen, Germany; [5]Department of Physics, Institute for Advanced Studies in Basic Sciences, Zanjan, Islamic Republic of Iran; [6]Aix-Marseille Université, CEA, CNRS, BIAM, F-13108, Saint-Paul-lez-Durance, France

*For correspondence:
stefan.klumpp@phys.uni-goettingen.de (SK);
damien.faivre@cea.fr (DF)

[†]These authors contributed equally to this work

Competing interests: The authors declare that no competing interests exist.

**Abstract** Bacteria propel and change direction by rotating long, helical filaments, called flagella. The number of flagella, their arrangement on the cell body and their sense of rotation hypothetically determine the locomotion characteristics of a species. The movement of the most rapid microorganisms has in particular remained unexplored because of additional experimental limitations. We show that magnetotactic cocci with two flagella bundles on one pole swim faster than 500 μm·s$^{-1}$ along a double helical path, making them one of the fastest natural microswimmers. We additionally reveal that the cells reorient in less than 5 ms, an order of magnitude faster than reported so far for any other bacteria. Using hydrodynamic modeling, we demonstrate that a mode where a pushing and a pulling bundle cooperate is the only possibility to enable both helical tracks and fast reorientations. The advantage of sheathed flagella bundles is the high rigidity, making high swimming speeds possible.

## Introduction

The understanding of microswimmer motility has implications ranging from the comprehension of phytoplankton migration to the autonomously acting microbots in medical scenarios (*Sengupta et al., 2017*; *Felfoul et al., 2016*). The most present microswimmers in our daily lives are bacteria, most of which use flagella for locomotion. Well-studied examples of swimming microorganisms include the peritrichous (several flagella all over the body surface) *Escherichia coli* with an occasionally distorted hydrodynamic flagella bundling (*Turner et al., 2000*) and the monotrichous (one polar flagellum) *Vibrio alginolyticus*, which are pushed or pulled by a flagellum and exploit a mechanical buckling instability to change direction (*Xie et al., 2011*; *Son et al., 2013*). The swimming speeds of so far studied cells are in the range of several 10 μm s$^{-1}$ and their reorientation events occur on the time scale of 50–100 ms (*Son et al., 2013*; *Berg and Brown, 1972*).

*Magnetococcus marinus* (MC-1) is a magnetotactic, spherical bacterium that is capable of swimming extremely fast (*Zhang et al., 2014*; *Fenchel and Thar, 2004*; *Bazylinski et al., 2013*; *Garcia-Pichel, 1989*). MC-1 as well as the closely related strain MO-1 (*Ruan et al., 2012*) are equipped with two bundles of flagella on one hemisphere (bilophotrichous cells). The bacterium also features a magnetosome chain, which imparts the cell with a magnetic moment ('magnetotactic' cell). They are

assumed to swim with the cell body in front of both flagella bundles, which synchronously push the cell forward (*Zhang et al., 2014*). This assumption leads to helical motion in the presence of a strong magnetic field, which exerts a torque on the cell's magnetic moment, as seen in hydrodynamic simulations (*Yang et al., 2012*). In the absence of a magnetic field, this model predicts rather straight trajectories.

Our observations disagree with the above-mentioned model, indicating that an understanding of the physics of their swimming is still missing, even though proof of concept biomedical applications of these bacteria have already emerged (*Felfoul et al., 2016*). Here, we confirm that MC-1 cells reach speeds of over 500 µm s$^{-1}$ (*Figure 1D*) and observe that the cells travel along a double helical path, which has not been reported for bilophotrichous cells so far. In addition, we observed that this rapid movement is complemented by an extremely fast reorientation ability (less than 5 ms). We connect the flagella bundle architecture and the swimming mechanism by hydrodynamic simulations and show that only a striking cooperative movement where one flagella bundle pushes while the other pulls the cell explains these motility characteristics.

## Results

MC-1 cells were observed in a physicochemically controlled environment that resembled the cell's natural habitat (*Adler, 1973*; *Figure 1A*). The cells were introduced into a flat, rectangular

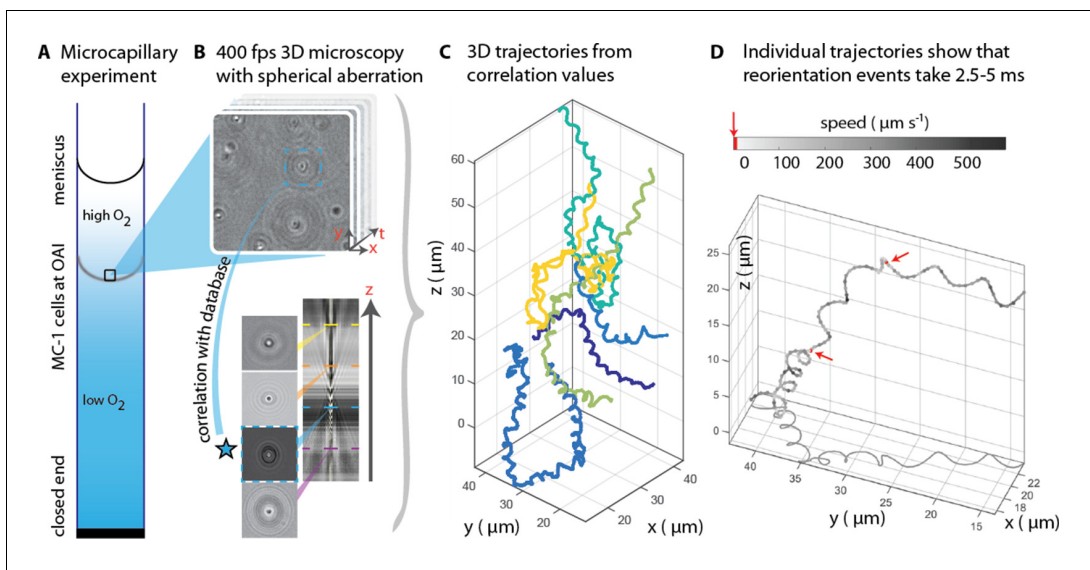

**Figure 1.** 3D tracking in physiochemically controlled conditions. (A) MC-1 cells were transitioned into agar-free medium, the solution was degassed to remove the oxygen and inserted into a microcapillary. Due to one open and one closed end, an oxygen gradient formed and the cells accumulated near their preferred microoxic conditions (the oxic-anoxic-interface, OAI). (B) The cells were observed near the band using 400 fps phase contrast video microscopy with a spherical aberration, which causes interference patterns around the spherical swimmers. These patterns can be correlated with patterns from silica beads of known height relative to the focal plane. (C) A tracking algorithm enables high-throughput 3D tracking of the microswimmers (*Taute et al., 2015*). Colors indicate different cells. (D) Individual tracks were analyzed and a clockwise helical travel path with a radius close to the cell diameter was found as well as instantaneous traveling speeds between 100 µm s$^{-1}$ and 500 µm s$^{-1}$. Tracks can be interrupted by rapid reorientation events that last only 2.5–5 ms. The helix parameters like pitch and period time do not change before and after an event, but apparently do so in the projected 2D tracks (see projected shadow in D).

The online version of this article includes the following figure supplement(s) for figure 1:

**Figure supplement 1.** Validation of event identification algorithm for 3D tracking on simulated tracks with known parameters.

**Figure supplement 2.** MC-1 swimming statistics.

**Figure supplement 3.** Oblique and top view of a typical event in an MC-1 swimming track.

microcapillary tube with a height of 200 μm, where they accumulated in a band near their preferred oxygen conditions. The cells were imaged near this band (*Frankel et al., 1997*; *Fenchel, 1994*) at central height to avoid surface interactions. The capillary was placed at the center of three orthogonal Helmholtz coils (*Bennet et al., 2014*), which were used to cancel the Earth's magnetic field with a precision of 0.2 μT after the band had formed. Hence, the cells' motion could be observed in the absence of significant magnetic torques. Tracking was performed in 3D at 400 frames s$^{-1}$ (fps) (*Figure 1B*). A high-throughput tracking method was used (*Taute et al., 2015*) (see Materials and methods) for the reconstruction of the tracks (*Figure 1C and D*).

Our first set of observations revealed that the cells traveled on helical paths (with a pitch of 5.3 μm ± 1.3 μm, a diameter of 1.7 μm ± 0.2 μm, and a period of 46 ms ± 32 ms, n = 65, errors are standard deviations) in the absence of magnetic torques (parameter extraction was performed with an automated track analysis algorithm, which was validated against simulated swim tracks, see *Figure 1—figure supplement 1* and *Figure 1—figure supplement 2*). In addition, abrupt changes of the direction of the helical axis were observed (e.g. around 90° in *Figure 1—figure supplement 3*). These directional changes occur within 2.5 ms to 5 ms, at least one order of magnitude faster than any previously analyzed reorientation events (*Son et al., 2013*; *Berg and Brown, 1972*). Directional changes did not occur via a continuous modulation of the ratio of radius and pitch (*Crenshaw and Edelstein-Keshet, 1993*), as it has been observed as a part of the chemotaxis of sperm (*Jikeli et al., 2015*). Rather, the helix parameters were the same before and after such an event. 3D tracking is essential to obtain this conclusion, as projected 2D tracks exhibit apparent changes in the helix parameters (see projected shadow of the track in *Figure 1D* and *Figure 1—figure supplement 3*).

The cells were further examined at 1640 fps in high-intensity dark-field video microscopy to visualize the cell body and the flagella bundle movement in detail (*Figure 2*, Materials and methods). A representative track of the cell body movement is shown in *Figure 2A* together with the tracked velocity. At such frame rates, a more complex movement pattern becomes apparent, which was not detectable during the 3D tracking at 400 fps. The cell track can be represented by a superposition of two helices, a small helix on a large helix (*Figure 2B*), resulting in a position over time

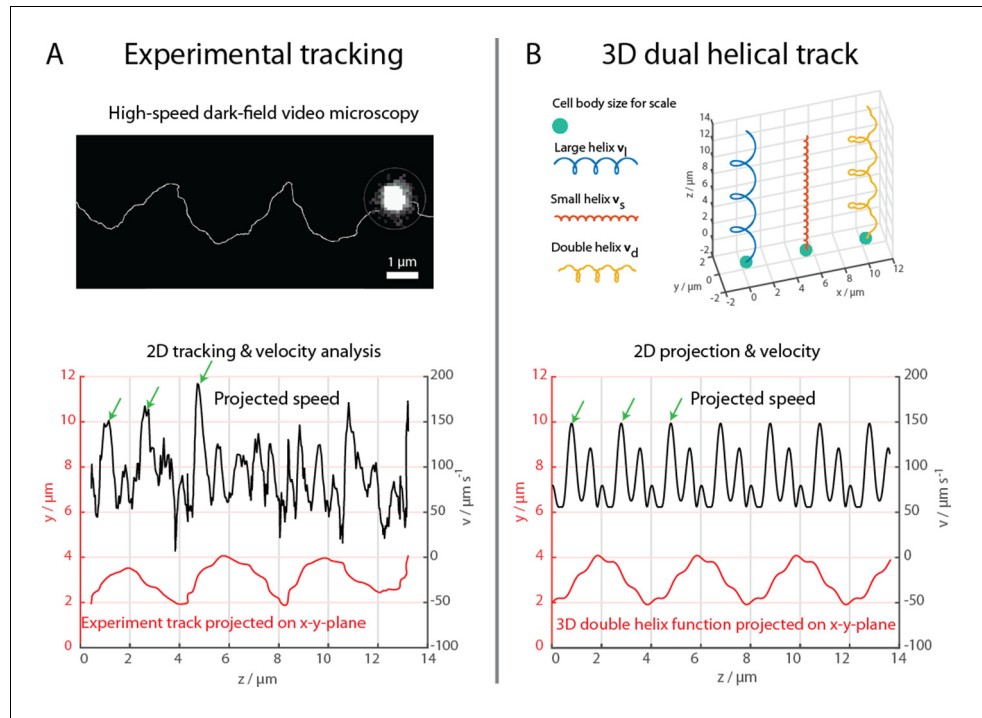

**Figure 2.** High-speed dark-field video microscopy at 1640 fps reveals a dual helical travel path of a MC-1 cell during free swimming. In (**A**) the tracked path is displayed after smoothing by a 5-point moving average filter and then plotted together with the cell velocity. Green arrows indicate velocity maxima. In (**B**) it is shown that the projected swimming path and projected velocity can be described by a projection of a large 3D double helix $\mathbf{v_d}(t)$.

$\boldsymbol{x}_d(t) = \left[ r_l \cos\frac{2\pi t}{T_l} + r_s \cos\frac{2\pi t}{T_s}, r_l \sin\frac{2\pi t}{T_l} + r_s \sin\frac{2\pi t}{T_s}, \; \frac{p_l t}{T_l} \right]'$. The large helical track featured a pitch $p_l$ of ~4 µm, a radius $r_l$ of ~1 µm and a period $T_l$ of 72 ms. The small helix featured a pitch of ~0.66 µm, a radius $r_s$ of ~0.125 µm and a period $T_s$ close to 14.4 ms. The ratio between this specific track's period times was close to 6, which was later used to choose the flagella bundle's rigidities and torque in the numerical simulations. The parameters for the small helix have been determined for a single track and do not possess statistical information. However, comparable double helical paths were apparent in all recorded tracks, including the cell track of the flagella imaging attempt (Figure 4, *Video 1* and *Video 2*).

The flagella bundle morphology and movements were imaged in transmission electron microscopy (TEM) and in high-intensity dark-field video microscopy (*Figure 3*, *Video 1* and *Video 2*). In the video microscopy at 1424 fps, short fibers next to the cell body and bright spots on the cell surface could be observed, which we identified as a part of a flagellum bundle close to the cell surface (*Son et al., 2013*). Although a state-of-the-art flagella imaging method, a sufficiently high framerate and a photon density close to the cell death limit were chosen, the resulting images reveal only little information about the exact flagella bundle positions and dynamics. This is the result of the combination of the small size of the cells and their extraordinary high swimming speed and possibly their strong flagella bundle movement.

Despite the difficulties in imaging the flagella bundles in full length, the position of the flagella bundle near the cell surface were tracked together with the cell's trajectory over 85 ms, which corresponds to 1.6 periods on the large helical trajectory of the cell (*Figure 4*). The observed movement pattern is more complex than previously assumed (*Felfoul et al., 2016*; *Frankel et al., 1997*; *Bazylinski et al., 2013*; *Waisbord et al., 2016*). The two flagella bundle positions moved rapidly around the cell. Crucially, one flagella bundle position is often seen in front of the cell (relative to its swimming direction), contrary to the model of two pushing flagella bundles, which remain behind the cell body. Additionally, the bright spots' movement pattern featured the same periodicity as the large helical swimming track of the cell (also compare *Figure 3C* and *Figure 3D*).

We turned to numerical simulations of the cell's swimming behavior, to develop a deeper understanding of the mechanisms of propulsion and rapid reorientation and to compensate the missing information from flagella bundle imaging. We performed Stokesian dynamics simulations (*Adhyapak and Stark, 2015*) for a spherical cell body (1.3 µm in diameter) with two discretized helical filaments (4 µm long and 50 nm thick if not stated differently) representing the flagella bundles (see Materials and methods for details). A large helical path of a microswimmer is produced from an off-axis (relative to the swimming direction) torque which continuously changes the direction of the thrust force. In case of a bilophotrichous cell,

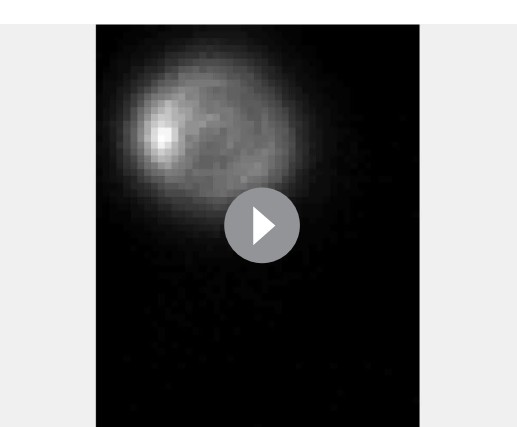

**Video 1.** Cell observation in a high-intensity dark-field video microscopy experiments at 1424 fps and 60x magnification with cell tracking.
https://elifesciences.org/articles/47551#video1

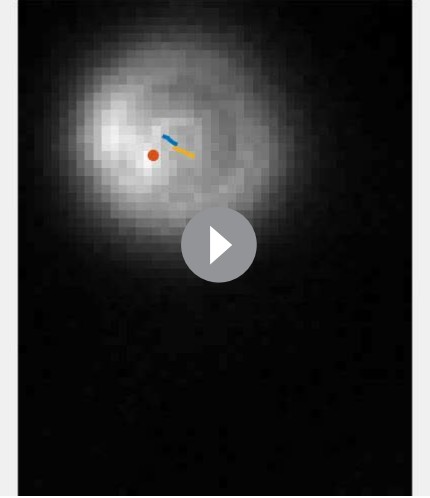

**Video 2.** Cell observation in a high-intensity dark-field video microscopy experiments at 1424 fps and 60x magnification with cell tracking and tracking of white spots on the cell surface.
https://elifesciences.org/articles/47551#video2

**Figure 3.** Flagella bundle imaging. (**A**) TEM image of an MC-1 cell stained with uranyl acetate. Two flagella bundles emerge from the MC-1 cell body. Each bundle features seven flagella, which emerge from a cavity on the cell surface. The individual flagella are bundled by a sheath (scale bar is 1 µm). (**B-D**) Individual frames from the high-speed dark-field video microscopy experiment at different points in time (*Video 1*, scale bars 1 µm). Bright spots appeared on the cell surface and next to the cell, which were identified as the parts of the flagella bundles that were closest to the cell surface. As the cell swam downwards in the image (blue track), the flagella bundle spots appeared in front of and behind the cell (relative to its overall movement direction). When comparing the times of B and C, the abrupt change in position indicates that another flagella bundle moved into focus of the microscope. When comparing C and D, it becomes apparent that the spots appear at nearly the same position on the cell at nearly the same horizontal position in the helical movement pattern, indicating a periodic flagella movement pattern that causes the helical track.

this requires a significant asymmetry in the propulsion force vectors of the two flagella bundles. Five possible asymmetries in flagella bundle configuration were considered to produce this torque and the resulting swimming paths were compared to experimental data (*Figure 5*). Three asymmetric configurations were ruled out numerically, as they did not result in a significant match between simulated and experimentally observed helix diameter, helix pitch and speed. These configurations are: A difference in flagella bundle length (*Figure 5—figure supplement 1*), a difference in motor strength (*Table 1*) and an asymmetry in the equilibrium angle of the two flagella bundles relative to the cell surface (*Shum, 2019*) (see Appendix 1). Another scenario would be a periodic, time-dependent movement of the flagella equilibrium angles relative to the cell surface (*Nguyen and Graham, 2017*). However, constant offsets in these angles already did not produce any significant matches between experiment and simulation. This indicates that this scenario will also fail this test and was not investigated further. Although not fully reaching experimentally observed helix pitches, the asymmetry in sense of motor rotation was the only scenario producing significant matches in helix diameter and cell speeds.

Since TEM images did not allow for a precise determination of the opening angle between the two flagella bundles (with respect to the body center; see, for example, *Figure 3A*), a wide range of opening angles (30˚−120˚) was used in the simulations (*Figure 5—figure supplement 2*). The effect of the flagellar motors was included as a torque at the base of the helices, which rotates the filament, and a counter-torque, which rotates the cell body. We considered the two bundles to rotate independently either counter-clockwise (CCW) or clockwise (CW). The parameters for bending rigidity, torsion rigidity and torque of the flagella bundles were adjusted to reproduce the observed movement characteristics (i.e. the parameters of the small helices and velocities). A motor torque of 12 pN µm, about 3.5 times the motor torque of an *E. coli* cell, had to be chosen together with high isotropic bending and twisting rigidities of 7 pN µm$^2$, about two times larger than for single flagella (*Adhyapak and Stark, 2015*). We assume that this increase is attributed to the structure of the flagella bundle, where seven protein motors power seven sheathed flagella cooperatively (*Figure 3A*). Only this assumption allowed for stable and high swimming velocities in the simulations, indicating that the function of the flagella bundle is to combine high torques with high rigidities.

Different swimming scenarios arising from combinations of CW and CCW rotation of the two bundles were simulated: both flagella bundles pushing the cell (CCW and CCW), both flagella bundles pulling (CW and CW) and one flagellum pushing, one pulling (CCW and CW). The fact that flagella can pull cells is well established (*Son et al., 2013*; *Constantino et al., 2016*) and our simulations

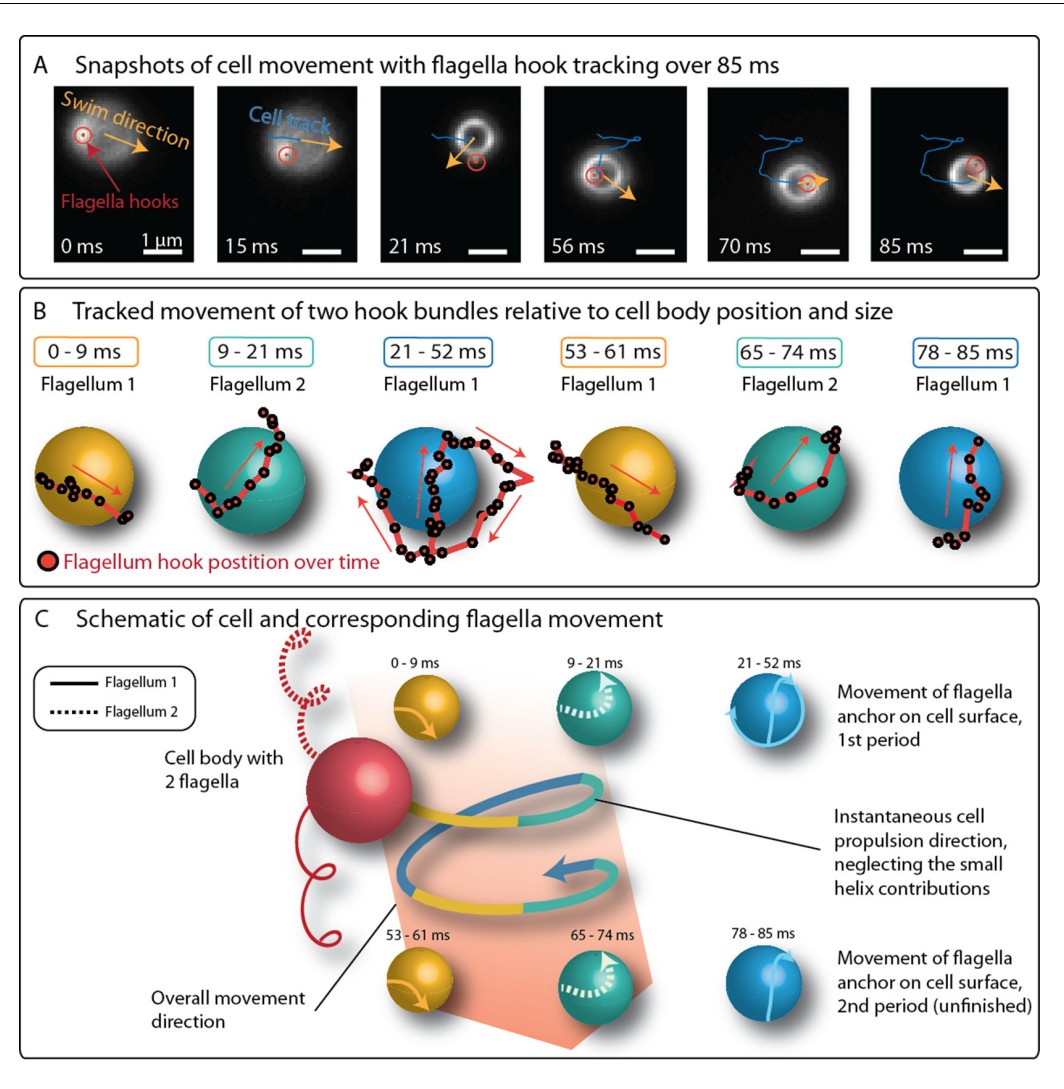

**Figure 4.** Positions of flagella bundles on the cell over 85 ms at 1424 fps. (**A**) The cell swam from top to bottom in the field of view. Moving bright spots were identified as the parts of the flagella bundles that were closes to the cell surface. The cell's helical traveling path had a period time of 52 ms. (**B**) The positions were tracked over time and are depicted relative to the center of the cell, represented by different spheres for different time intervals. (**C**) The schematic of the cell's large helical track is color-coded according to the different time intervals where the flagella bundle parts were visible to highlight the periodicity of the flagella bundle movement.

show that only the CCW and CW model results in the double-helical tracks experimentally observed (*Figure 4A and B*, *Figure 5—figure supplement 2*, *Video 3* and *Video 4*). Tracking the simulated cells resulted in time traces of the velocity and the projected position that are strikingly similar to the experimental ones (for a quantitative comparison, see *Figure 5* and *Figure 2A*). We tested the dependence of the trajectories on model parameters, in particular the opening angle and motor torque (*Table 2* and *Table 3*). Double helical trajectories were observed for flagellar opening angles in the whole parameter range, with diameter and speed in agreement with the experiments (but no fully quantitative agreement for the pitch of the large helix).

We also tracked the position of the flagella bundle positions in the simulations (by tracking the first beads of each discretized flagellum). Both flagella bundles rotated with 100 Hz - 150 Hz and one period of the rotation coincides with one period of the large helix in the cell's trajectory. This agrees with the flagella bundle movement in the experiments (*Figure 4*).

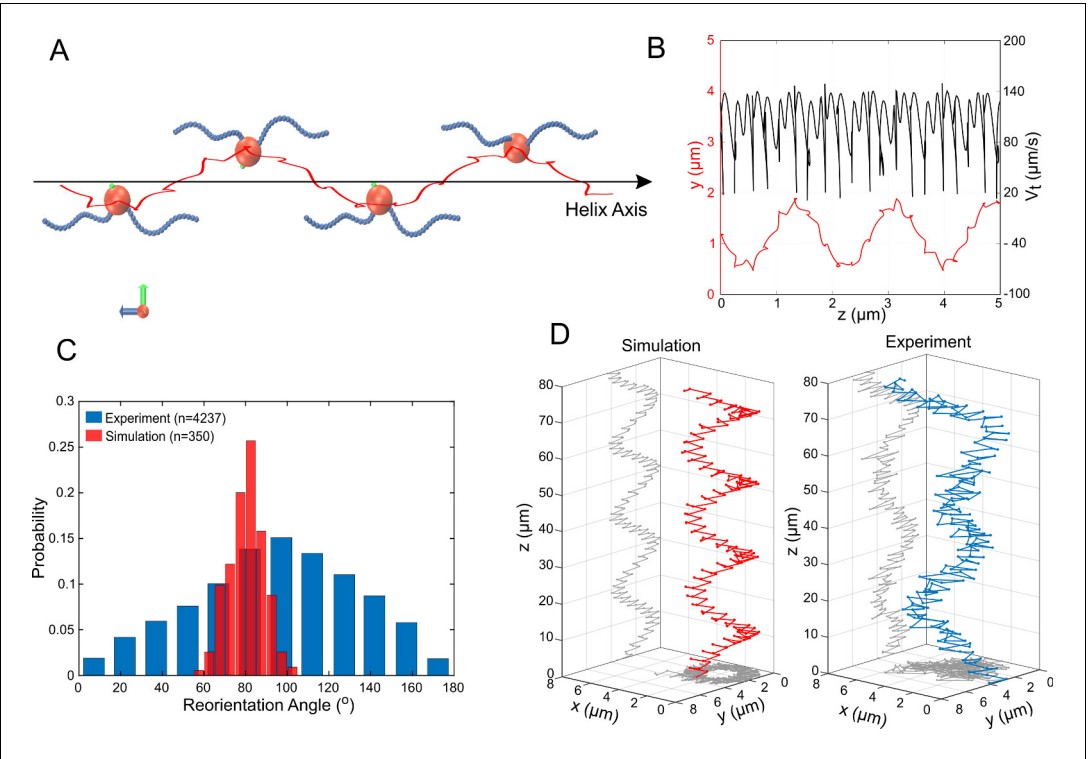

**Figure 5.** Simulations of MC-1 swimming dynamics with one pushing and one pulling flagellar bundle. The principle cell and flagella bundle arrangement is shown in (**A**) at four distinct time points where the cell body diverges strongly from the helix axis. (**B**) Shows the projected track of a simulated MC-1 with the flagellar opening angle 60° together with the projected speed. The results are comparable to measured data from *Figure 2A*. (**C**) Histogram of turning angles for the reorientation events seen in experiments and in simulations, where reorientation results from periods of synchronous rotation. (**D**) Validation of the cooperative pushing and pulling model in the presence of a strong magnetic field (3 mT). A third hyper-helix is observed in both experiment and simulation.

The online version of this article includes the following figure supplement(s) for figure 5:

**Figure supplement 1.** Swimming path parameters from numerical simulations with an asymmetry in flagella lengths compared to experimental values.

**Figure supplement 2.** Comparison of cell path parameters from experiments and the pushper-puller simulation scenario.

**Figure supplement 3.** Reorientation angle histogram for the following transient flagella bundle configuration: CCW and CW → CCW and 0 → CCW and CW (the rotation of the puller flagella is stopped during reorientation).

**Figure supplement 4.** Reorientation angle histogram for the following transient flagella bundle configuration: CCW and CW → CCW and CCW → CW and CCW → CW and CW → CCW and CW.

**Figure supplement 5.** Reorientation angle histogram for the following transient flagella bundle configuration: CCW and CW → CCW and CCW → CW and CCW → CCW and CCW → CCW and CW.

**Figure supplement 6.** Simulations of different scenarios of MC-1 swimming with varying magnetic fields and magnetic moment orientation.

Our model suggests a pushing flagella bundle together with a pulling flagella bundle rotating in opposite senses. The experimentally observed rapid reorientation events that lasts between 2.5 ms and 5 ms are most reasonably produced by a sharp change in the rotation of at least one flagella bundle. We tested different scenarios (*Figure 5—figure supplements 3–5*) and found a transiently synchronous rotations of the two flagella bundles to be the most likely process. We tested this by changing the sense of rotation from CCW and CW to CCW and CCW for 4 ms in the simulations and calculating the angle between the trajectory segments before starting and after finishing this transient CCW and CCW step (outtake in *Video 5*). This procedure indeed resulted in rapid reorientation with a change in direction by 80° ± 8° (*Figure 4C*), in agreement with the 94° ± 39° change seen

**Table 1.** Simulated swim track parameters for an asymmetry in motor torques for MC-1 cells for different opening angles between the two flagella bundles.
The second row gives statistical averages of experimental values for diameter (D), pitch (P), period time (T) and speed ($V_t$).

| Opening angle | Tm2/Tm1 | D (µm) | P (µm) | T (ms) | $V_t$ (µm/s) |
|---|---|---|---|---|---|
| | Experiment | 1.7 | 5.3 | 46.0 | 100.0 |
| 60 ˚ | 0.1 | 0.43 | 0.93 | 47.4 | 52.33 |
| | 0.5 | 0.29 | 0.73 | 20.2 | 58.85 |
| | 0.9 | 0.18 | 1.58 | 33.4 | 88.42 |
| 80 ˚ | 0.1 | 0.54 | 1.08 | 54.4 | 54.76 |
| | 0.5 | 0.39 | 1.09 | 32 | 60.44 |
| | 0.9 | 0.20 | 1.61 | 34.8 | 84.14 |

in the experiments (errors are standard deviations). The standard deviation of the directional change is small in our simulations, where only the runtime was varied, compared to the experimental value. The mismatch likely arises from biological diversity in flagella lengths and opening angles, but also due to shifts in local physiochemical conditions, which can for example influence the motor torque (*Son et al., 2013*). A transient buckling deformation was observed in the simulated flagella bundles during an event, which caused the fast reorientations.

To further validate our simulations, we predicted the cell's swimming behavior at high magnetic fields using our simulation without changing further parameters. The direction of the cell's magnetic moment was assumed to be perpendicular to the bisector of the two flagella bundle axes (further scenarios can be found in *Figure 5—figure supplement 6*). The simulations showed an additional, large hyper-helical movement pattern (with diameter $D_{sim} \simeq 3.9$ µm and pitch $P_{sim} \simeq 19.1$ µm). The same pattern could thereafter be found in experimental data with similar parameters ($D_{exp} \simeq 4.2$ µm and $P_{exp} \simeq 30$ µm). Typical MC-1 hyper-helical trajectories from simulations and experiments are shown in Figure 5D.

## Discussion

Magnetococci are exceptional swimmers with respect to both their high speed and their reorientation swiftness. In addition, they are likely to make use of a previously unrecognized pattern of motion of their flagella bundles, with one bundle pushing the cell body and the other pulling it. Key to observing this pattern of motion was the 3D tracking of single cells in the absence of magnetic torques to observe the unexpected two-helix trajectories, together with hydrodynamic simulations.

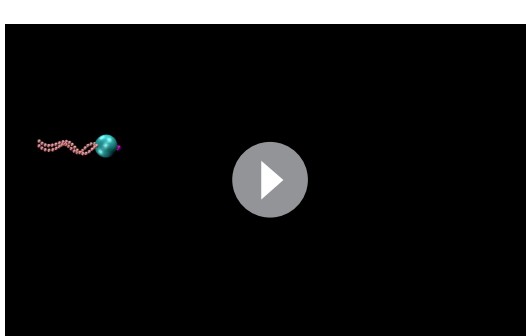

**Video 3.** Simulation of cell swimming with two pushing flagella.
https://elifesciences.org/articles/47551#video3

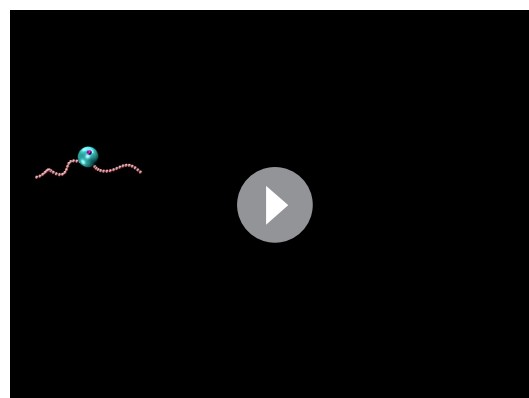

**Video 4.** Simulation of cell swimming with one pulling and one pushing flagella.
https://elifesciences.org/articles/47551#video4

The hierarchically organized flagella bundles provide a high torque and rigidity, necessary to reach record speeds of over 200 body lengths per second, while the unusual type of coordination of the two bundles provides a mechanism for rapid reorientation. Magnetotactic cocci such as MC-1 thrive and generally represent the most abundant MTB in aquatic environments (*Lefèvre et al., 2015*). Their unusual motility is certainly an adaptation that represents a selective advantage that make them competitive in the highly coveted biotopes that are the oxic-anoxic interfaces (*Brune et al., 2000*). Indeed, the observed geometry of this resulting swimming pattern is reminiscent of the situation recently reported for *magnetospirilla* (*Murat et al., 2015*), despite the difference in body plan. The *magnetosprilla* have two flagella at opposite cell poles, and a magnetic moment parallel to the flagellar axis, while the flagella of the cocci studied here are attached on one hemisphere of the cell and almost perpendicular to the magnetic moment. Nevertheless, both swim aligned with a magnetic field with one flagellum ahead and one trailing. In summary, record-breaking cells like MC-1 can help to understand physical limits of natural microswimmers and provide design principles for their artificial counterparts.

## Materials and methods

### Key resources table

| Reagent type (species) or resource | Designation | Source or reference | Identifiers | Additional information |
|---|---|---|---|---|
| Cell line (*Magnetococcus marinus*) | MC-1 | CEA | NCBI:txid156889 | Dr. Christopher Lefèvre, CNRS |
| Software, algorithm | ImageJ | NIH | RRID:SCR_003070 | |
| Software, algorithm | MatLab | The MathWorks | RRID:SCR_001622 | |

### Cell medium and culturing

MC-1 was cultured similarly to the procedure reported by *Bazylinski et al. (2013)*. Artificial sea water (ASW) was used as a base medium, containing 20 g NaCl, 6 g $MgCl_2$, 2.4 g $Na_2SO_4$, 0.5 g KCl and 1 g $CaCl_2$ per liter $H_20$. To this was added (per liter) the following, in order, prior to autoclaving: 0.05 mL 0.2% (w/w) aqueous resazurin, 5 mL Wolfe's mineral solution (ATCC, MD-TMS), 0.3 g $NH_4Cl$, 2.4 g HEPES and 1.6 g agar (Kobe I, Carl Roth). The medium was then adjusted to pH 6.3 and autoclaved. After the medium had cooled to about 45°C, the following solutions were added (per liter), in order, from previously sterile-filtered stock solutions: 0.5 ml vitamin solution (ATCC, MD-VS-10mL), 1.8 mL 0.5 M potassium phosphate buffer, pH 7, 3 mL0.01 M $FeCl_2$ and 40% (w/w) Na thiosulfate. Finally, 0.4 g cysteine was added (per liter), which was made fresh and filter-sterilized indirectly into the medium. The medium (12 mL) was dispensed into sterile Hungate tubes after verifying a pH of 7.0. All cultures were incubated at room temperature (~25°C) and, after approximately one week, a microaerobic band of MC-1 formed at the oxic–anoxic interface (pink-colorless interface) of the tubes. The cells were harvested in volumes of 1 mL from that region and magnetically transferred to ASW for experiments. The transfer step was necessary to remove agar for swimming experiments and to minimize background scattering in dark-field microscopy.

### Cell morphology analysis

The flagella bundle length was determined with ImageJ from images taken with a Zeiss EM 912 Omega transmission electron microscope using an acceleration voltage of 120 kV. The cells were dried on a carbon film on a regular TEM copper grid and stained with 4% uranyl acetate for 6 min. Due to the staining, the cell walls appeared electron dense and covered the sight on flagella on top or below the cells. Hence, we added the average cell radius to the mean of the flagella length. A mean flagella bundle length of 3.3 μm ± 0.4 μm (n = 27) resulted. The size of the non-dried cells were measured with ImageJ from images taken with a LSM780 (Zeiss; Germany) confocal microscope. The mean size was 1.3 μm ± 0.1 μm (n = 103).

### Microcapillary experiments

1 mL of a freshly harvested sample was degassed using nitrogen for 15 min and the sample was introduced into a rectangular micro-capillary (VitroTubes, #3520–050,) by capillary forces. One end

**Table 2.** Swimming features of MC-1 cells using CCW and CW swimming mechanism for simulations with different flagellar opening angles.

The given output parameters are the helix diameter, its pitch, the period time (period), the effective velocity ($V_z$) and the instantaneous velocity ($V_t$).

| Flagellar opening angle | Diameter (µm) | Pitch (µm) | Period (ms) | $V_z$ (µm/s) | $V_t$ (µm/s) |
|---|---|---|---|---|---|
| 30° | 2.1 | 3.0 | 144 | 21 | 76 |
| 45° | 1.7 | 2.3 | 88 | 27 | 87 |
| 60° | 1.4 | 1.7 | 59 | 30 | 96 |
| 80° | 1.1 | 1.2 | 39 | 32 | 106 |
| 100° | 0.9 | 1.1 | 34 | 32 | 108 |
| 120° | 0.7 | 1.1 | 34 | 33 | 106 |

of the capillary was sealed with petroleum jelly and the capillary was mounted on a microscope slide that was used to hold the sample on the microscope stage. The oxygen diffusion from the open end caused an oxygen gradient inside the medium, which led together with the oxygen consumption of the cells to the formation of a microaerobic bacteria band. The band formed in the presence of a 50 µT magnetic field towards the sealed end. The tracks were taken after 30 min of microcapillary infiltration at 0 µT.

### 3d tracking experiments

3D swim tracks were recorded at 400 fps in a microcapillary in the vicinity of the microaerobic band (Nikon, S Plan Fluor ELWD, ×40, Ph2, NA 0.6; NA 0.76 condenser lens; Ph2 aperture ring, 635 nm LED illumination). The 3D tracks were reconstructed using the high-throughput phase contrast reference method by *Taute et al. (2015)*. A spherical aberration was introduced using a misalignment of the correction collar of the ×40 phase contrast objective to a cover slip thickness correction of 1.2 mm. The aberration caused inference pattern, which can be correlated with the relative height of the microswimmer. A custom made microscope platform, developed by *Bennet et al. (2014)*, was used, which features three orthogonal Helmholtz coil pairs around the sample position. The setup can generate homogeneous fields at the sample position with arbitrary direction with a precision of 0.2 µT. The Earth's magnetic field was canceled or an artificial field towards low oxygen conditions was generated during a capillary experiments.

### Dark-field microscopy

Flagella bundle positions were visualized at 1424 fps using high-intensity dark-field microscopy (Nikon 60×, 0.5–1.25 NA CFI P-Fluor oil objective at 0.75 NA; 1.2 NA oil condenser; mercury lamp illumination) and an Andor Zyla 5.5 (10-tap) camera (6.5 µm per pixel). The cells were placed inside a 10 µm deep chamber in ASW. A deeper chamber did not allow for successful dark-field imaging of flagella bundles due to an increase in noise from background scattering. The focal plane was adjusted to the center of the chamber, such that interactions between the observed flagella bundles and the chamber surfaces were avoided. Presumably due to the flagella bundle size and the high

**Table 3.** Swimming features of MC-1 cells for different motor torques.

The 3.5 times increase of motor torque compared to *E-coli* cells was chosen for the simulations described in the main text due to the best fits of swimming track parameters.

| Motor torque ($T_m/T_{m\text{-Ecoli}}$) | D (µm) | P (µm) | T (ms) | $V_t$ (µm/s) |
|---|---|---|---|---|
| 3.5 | 1.4 | 1.7 | 59 | 96 |
| 3 | 1.44 | 1.56 | 74.4 | 80.84 |
| 2.5 | 1.43 | 1.64 | 84.2 | 69.41 |
| 2 | 1.49 | 1.39 | 101.2 | 53.27 |

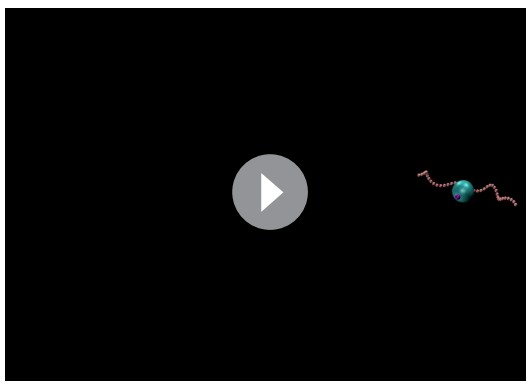

**Video 5.** Example simulation of reorientation events.
https://elifesciences.org/articles/47551#video5

rotation speed of the cell and of the bundles, a direct observation of the whole flagella bundles was not successful and only a small section of each flagellum could be visualized. A sub-millisecond exposure time set the requirement for high photon intensity at the sample position. The intensity was increased until the cells melted instantly when swimming into focus. A green filter prevented the melting while still facilitating sufficient brightness. Scrupulous cleanliness at all optical interfaces was obligatory. High-speed cell body tracking could be accomplished at the center of a 200 μm deep chamber. Optimized visualization was achieved at 1640 fps using dark-field microscopy without the critical illumination from flagella tracking (Zeiss 60×, 1.0 NA; 1.2 NA oil condenser; halogen lamp illumination). The dark-field setup did not allow for a cancellation of external magnetic fields. Measurements of the magnetic field at the capillary position yielded $B_x = -195$ μT ± 0.2 μT, $B_y = 60$ μT ± 0.2 μT and $B_z = 27$ μT ± 0.2 μT.

## Cell and flagella tracking software

3D track reconstruction was performed in MatLab (The MathWorks) using an adapted version of the code from *Taute et al. (2015)*. For automated analysis of the reorientation angle distribution, the script was successfully tested against simulated data with known reorientation angles (See SI). The helix parameter determination was performed automatically on tracks extending a duration of 0.4 s, with a mean-square-displacement of at least 10 μm$^2$ and a mean opening angle between all consecutive velocity vectors of less than 60 ° to exclude strongly irregular tracks. The same exclusion parameters have been used for the reorientation angle analysis. Simultaneous dark-field cell and flagella tracking was performed using an in-house semi-automatic MatLab program. Tracking of the cell trajectory at 1640 fps without the flagella movement was performed using the TrackMate plugin of ImageJ.

## Hydrodynamic simulations

Each flagellum was modeled as a helical filament and a rotary motor. The helical filament was discretized with 20 beads with a discretization distance of 200 nm and bead diameter of 50 nm. Excluded volume interactions between all particles are considered using a truncated Lennard-Jones potential. Hydrodynamic interactions are taking into account using Stokesian dynamics simulation method having the translational anisotropic friction coefficients of $\gamma_{\text{Noentity}} = 1.6 \cdot 10^{-3}$ pNs/μm$^2$ and $\gamma_\perp = 2.8 \ 10^{-3}$ pNs/μm$^2$ and the rotational friction of $\gamma_r = 1.26 \ 10^{-6}$ pNs for the flagellum beads and $\gamma_{bt} = 6\pi\eta R_b$ and $\gamma_{br} = 8\pi\eta R_b^3$ for the translational and rotational friction coefficients of the cell body. Irrespective of the high motor torque and isotropic bending and twisting rigidities mentioned in the main text, a stretching rigidity of 1000 pN, comparable to that of single flagella, could be chosen. A Rotne-Prager matrix was used for calculating the cross-mobilities and cross-hydrodynamics (*Dhont, 1996*). The swimming dynamics of the model cell at low Reynold number was calculated by solving the translational and rotational Stokes equations of motion for the cell body, flagellar beads and the bonds between them. A second-order Runge-Kutta algorithm (*Sewell, 1988*; *Press et al., 1992*) and simulation time-steps of $10^{-7}$ s were used to solve the equations of motion numerically.

## Acknowledgements

The research leading to these results was supported by the Max Planck Society and by Deutsche Forschungsgemeinschaft (DFG) within the priority program on microswimmers (grants No. KL 818/2–2 and FA 835/7–2 to SK and DF). Further, SM was supported by Deutscher Akademischer Austauschdienst, DAAD (grant no. 57314018) as well as Deutsche Forschungsgemeinschaft (DFG) through SFB

937. AC is funded by the IMPRS on Multiscale Biosystems. CTL acknowledges support by the French National Research Agency (ANR Tremplin-ERC: ANR-16-TERC-0025–01). The authors would further like to thank L Alvarez, R Pascal and J Jikeli for measurement trainings, C Oschatz for medium preparation, A Pohl for electron microscopy and C Pilz for laboratory assistance.

## Additional information

### Funding

| Funder | Grant reference number | Author |
|---|---|---|
| Max-Planck-Gesellschaft | | Damien Faivre |
| Deutsche Forschungsge-meinschaft | FA 835/7-2 | Damien Faivre |
| Deutsche Forschungsge-meinschaft | KL 818/2-2 | Stefan Klumpp |
| Deutscher Akademischer Aus-tauschdienst | 57314018 | Sarah Mohammadinejad |
| Deutsche Forschungsge-meinschaft | SFB 937 (A21) | Stefan Klumpp |
| Agence Nationale de la Re-cherche | ANR-16-TERC-0025-01 | Christopher T Lefèvre |
| IMPRS on Multiscale Biosys-tems | | Agnese Codutti |
| French National Research Agency | ANR Tremplin-ERC: ANR-16-TERC-0025-01 | Christopher T Lefèvre |

The funders had no role in study design, data collection and interpretation, or the decision to submit the work for publication.

### Author contributions

Klaas Bente, Conceptualization, Data curation, Software, Formal analysis, Validation, Investigation, Visualization, Methodology; Sarah Mohammadinejad, Data curation, Software, Formal analysis, Validation, Investigation, Visualization, Methodology; Mohammad Avalin Charsooghi, Software, Validation, Investigation, Visualization; Felix Bachmann, Software, Methodology; Agnese Codutti, Software, Validation, Methodology; Christopher T Lefèvre, Resources, Supervision, Methodology; Stefan Klumpp, Conceptualization, Resources, Formal analysis, Supervision, Funding acquisition, Methodology; Damien Faivre, Conceptualization, Resources, Supervision, Funding acquisition, Validation, Project administration

### Author ORCIDs

Klaas Bente (ID) https://orcid.org/0000-0002-2520-4697
Sarah Mohammadinejad (ID) https://orcid.org/0000-0002-3758-5693
Mohammad Avalin Charsooghi (ID) https://orcid.org/0000-0002-7772-8513
Stefan Klumpp (ID) https://orcid.org/0000-0003-0584-2146
Damien Faivre (ID) https://orcid.org/0000-0001-6191-3389

### Decision letter and Author response

Decision letter https://doi.org/10.7554/eLife.47551.sa1
Author response https://doi.org/10.7554/eLife.47551.sa2

## Additional files

### Supplementary files

• Transparent reporting form

## Data availability

3D tracks have been deposited in Dryad Digital Repository (https://doi.org/10.5061/dryad.r2nd550).

The following dataset was generated:

| Author(s) | Year | Dataset title | Dataset URL | Database and Identifier |
|---|---|---|---|---|
| Bente K, Moham-madinejad S, Char-sooghi MA, Bachmann F, Co-dutti A, Lefèvre CT, Klumpp S, Faivre D | 2019 | Data from: High-speed motility originates from cooperatively pushing and pulling flagella bundles in bilophotrichous bacteria | https://doi.org/10.5061/dryad.r2nd550 | Dryad Digital Repository, 10.5061/dryad.r2nd550 |

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

## Appendix 1

## Supplementary Information

### Rapid change in swimming direction

An exemplary track of a fast reorientation event is given in *Figure 1—figure supplement 3*. A change in swimming direction by roughly 90 ˚ occurred in less than 5 ms.

### 3D tracking validation

The algorithm for 3D track event detection was validated using Langevin-simulations of active Brownian particles with defined mean reorientation angles, velocities, mean run times and mean run lengths, as described in *Figure 1—figure supplement 1*. Two parameter sets have been simulated and used for validation. The evaluation shows good agreement between input and output parameters. The swim track parameters that have been determined from 3D tracks of MC-1 cells in the capillary experiment are presented in *Figure 1—figure supplement 2*.

### Relation between helical and effective travel path

The overall traveled path l on a helix during one period can be parameterized using radius and pitch by $l = \sqrt{p^2 + 4\pi^2 r^2} = 7.5$ *m*. While effectively traveling 4 μm in 72 ms, the cell moves 7.5 μm on the large helix. While the contribution of the small helix to the actual swimming path provides another factor of 1.1, the ratio between instantaneous speed and effective speed is near 2.

## Test of different asymmetries in flagella configurations

### 1) Asymmetry in senses of rotation of motors

This asymmetry is described in the main text.

### 2) Asymmetry in flagella lengths

Simulations for different combinations of flagella lengths, L1 = 2, 3 μm, L2 = 4, 5, 6, 7, 8 μm (with 2 opening angles 60, 80) were performed. With this asymmetry in flagella length, a helical path can be observed but, as shown in *Figure 5—figure supplement 1*, the size of the second helix isn't comparable to the experimental values.

Moreover, the curves in *Figure 5—figure supplement 1* are flat, indicating that changing the strength of asymmetry doesn't have remarkable effect on the size of the second helix.

### 3) Asymmetry in motor strengths

In these simulations the motor torque of one flagellum (Tm2) is 0.1, 0.5 or 0.9 times the torque of the other flagellum (Tm1). We observed double-helical trajectory with this motor asymmetry as well but with considerably smaller size compared to the experiment and our proposed pusher-puller model. The size of helix and swimming features are presented in *Table 1*, as well as experimental values for comparison.

Among these simulations the largest helix is observed for Tm2/Tm1 = 0.1 which is still smaller than the size of the experimental helix. Moreover, a hypothetical multi-flagellated microorganism that features flagella with a huge difference in motor torque (Tm2/Tm1 = 0.1) it is biologically not relevant. Also, the simulated velocities did not fit the experimental values for any of the tested scenarios.

### 4) Asymmetry in flagella direction

We performed simulations with asymmetry in flagella direction. For this purpose, combination of flagella angles 0° and 10°, 0° and 20°, 0° and 30° and 0° and 40° as hinge equilibrium angles have been tested. In swimming trajectories, a very weak second helix with diameters of about an order of magnitude smaller than our pusher-puller model was observed.

For example, for 0° and 30° resulted in D = 0.17 μm, p=3.18 μm, $V_t$ = 71.50 μm/s. Which is far from the experimental ones: D = 1.7 μm, p=5.3 μm, $V_t$ = 100.0 μm/s.

In these simulations, the more tilted flagella bundle is only tilted close to the cell surface, but due to flexibility, the tilt is not persistence along the length of the flagella bundle and the two bundles approach each other at their end parts. So, the results show that asymmetry in flagella direction cannot generate a helix comparable to MC-1's large helix.'

## Discussion of pitch differences between simulation and experimental observation

While the pitch of the simulation result, presented in the main text, is off by a factor of 2, the diameter and period time of helical trajectories are in good agreement with the experiments. The simulations show that the features of helical trajectories strongly depend on the flagellar opening angle (see *Table 2*) such that increasing the flagellar opening angle decreases the pitch and diameter of the large helix. Simulated effective velocities are smaller than the tracked velocities by factor of 2, since the pitch is decreased by the same factor while the period times are comparable.

Since the simulations with constant flagella length did not lead to a satisfying match in pitches, further simulations for different combinations of opening angles and flagellum lengths were carried out. The resulted diameter, pitch and velocity are compared with experimental values in *Figure 5—figure supplement 2*.

From these simulations, we can conclude that there are some other combinations of opening angle and flagellum length (for example L = 5 μm, opening angle = 60°) with which we can achieve a similar qualitative matching between simulation and experiment. However, none of them satisfy all the parameters simultaneously. The best match between helix diameter, pitch and speed is described in the main text. However, we clarify here that other tuning parameters (for example flagellum length and diameter, flagella opening angle, the motor torque) in the model cannot be precisely determined from TEM or optical microscopy images of MC-1. Therefore, it is computationally very time consuming to test different values for all these tuning parameters to find an accurate quantitative match for all parameters between experiment and simulation.

We further investigated several motor torques. Due to the collapse of our simulation for strong motors (because of the arising numerical error), the maximum applicable torque was 12 pN μm (about 3.5 times the motor torque of an *E. coli*). However, we can extrapolate the swimming behavior of our model MC-1 at higher motor torques by looking at its trend for lower values of motor torque. We tried motor torques of 2, 2.5, 3, and 3.5 times the motor torque of *E. coli* and the results are listed in *Table 3*.

The top row represents the result discussed in the main text. The overall pitch increases with increasing motor torque. It can be concluded that the motor torque can be one of the parameters that may help matching between experiment and simulation, although the definitive match was not reached in this study.

## Transient flagella configurations that produce fast reorientation events

Three possible transient flagella configurations for reorientation events were tested in the simulations (CCW meaning counter-clockwise and CW meaning clockwise).

1. CCW and CW → CCW and 0 → CCW and CW (the rotation of the puller flagella is stopped during reorientation). An average reorientation angle of 25° resulted.

2. CCW and CW → CCW and CCW → CW and CCW → CW and CW → CCW and CW. The average reorientation angle for this transient configuration change was 105°.
3. CCW and CW → CCW and CCW → CW and CCW → CCW and CCW → CCW and CW. The average reorientation angle for this transient configuration change was 105°.

Based on these results, we concluded that the reorientation angle statics produced by CCW and CW → CCW and CCW → CCW and CW matched the best to the experiment result.

## Flagella bundle morphology

The MC-1 cell shape is given in *Figure 3A* in the main text, where seven individual flagella were identified, emerging from a sunken pit, as already described in *Bazylinski et al. (2013)*. The individual bundles of the close relative MO-1 (*Ruan et al., 2012*) contain seven individual flagella, as well, which emerge from a hexagonal pattern on the cell surface. Additionally, 24 gap-filling, presumably friction-reducing microfibrils were found in *Ruan et al. (2012)*. We assumed a similar flagella arrangement in this study, leading to a cooperative torque generation of the individual flagella in one sheath of an MC-1 cell.

## Test of different motor torque, magnetic moment direction and magnetic field configurations

To check for the validity of simulation prediction, an experiment is set up in low and high magnetic fields, 50 $\mu T$ and 3 mT, respectively. Over 1000 tracks are extracted and investigated for each magnetic field. For more certainty, swimming trajectories are investigated both in oxic and anoxic regions. In high magnetic field, over 80 trajectories can be detected illustrating the hyper-helix, while in Earth magnetic field only a few trajectories with semi-hyper-helix can be observed. A typical MC-1 trajectory with the mentioned hyper-helix resulted from simulation and experiment is shown in *Figure 4D*. A measured diameter and pitch of the hyper-helix, $D_{\text{exp}} \simeq 4.2 \mu m$ and $P_{\text{exp}} \simeq 30 \mu m$, are in reasonable agreement with the simulated one, $D_{\text{sim}} \simeq 3.9 \mu m$ and $P_{\text{sim}} \simeq 19.1 \mu m$.

