## [Decision Letter]

**Acceptance summary:**

This paper reports on the discovery of an unusual and fascinating form of motility in a magnetotactic bacterium. Through a combination of experiment and theory the authors have found that the two flagellar bundles work in opposite ways – one pushing and one pulling the cells through their fluid environment in double-helical paths. The very fast motility and very rapid trajectories changes appear to arise from these features. This work will surely be of interest not only to those interested in the biology of motility, but also physical scientists interested in the fluid dynamics of locomotion.

**Decision letter after peer review:**

Thank you for sending your article entitled "High-speed motility originates from cooperatively pushing and pulling flagella bundles in bilophotrichous bacteria" for peer review at *eLife*. Your article is being evaluated by three peer reviewers, and the evaluation is being overseen by a Reviewing Editor and Detlef Weigel as the Senior Editor.

The essence of the criticisms concerns the interpretation of the experimental observations (and the possibility of achieving better imaging of the flagella) and the analysis of the computational results. In both cases further details are needed, and in the former it appears that additional imaging would be in order.

Reviewer #1:

In my opinion the topic of the article is interesting as it identifies a new flagellar arrangement for motility (though similar simultaneous pushing/pulling has been observed for species with bundles at opposite poles rather than the same hemisphere, both in magnetospirilla). However, I am not completely convinced by the authors' evidence. As detailed below, while the story is quite plausible, the experiments seem to not have key evidence which I would have expected to have easily been observed if the scenario were true, and the numerics are not performed (or perhaps just not described) in a way that convincingly rules out other hypotheses.

1) In the dark field imaging, bright spots on the cell body are identified with flagella. This identification is not obvious to me, and it is quite puzzling why the authors do not actually image the flagellar bundles. If the bundles were indeed extended from the body in front and behind the cell, I would expect that they could be visualized which would definitively prove their proposed configuration. In Son, Guasto and Stocker, 2013, sheathed flagella are quite readily imaged. In my experience, there may be difficulty seeing flagella with darkfield at 1640 fps, but they should be readily imaged at <200 fps using their camera. Note that 200 fps is more than fast enough to resolve the larger helical motion with period 72 ms. It is unclear if the authors attempted this. If the authors tried and were not able to see a leading and lagging bundle, I would take that as evidence against their claim.

2) The authors describe the double helical path as "unexplored" in the Introduction, but based on my knowledge, double helical paths are what should be expected from most types of propulsion by flagellar bundles, with the main distinguishing feature being the size of the larger helix. The smaller helix has in the past been attributed to the rotation of the bundle or flagellar helix, see Keller and Rubinow, 1976. The larger helix arises anytime there is non-axisymmetric propulsion, see Hyon, Powers, Stocker, Fu 2012 and has been observed in many species, albeit with varying sizes of the larger helix. As discussed in point 3 below, an important implication of this is that the bar for numerical simulations cannot be simply qualitative as in "straight" vs. "double-helical," but must involve some quantitative analysis of the trajectory pitch and radius.

3) The numerics have not been described in a way that clearly rules out alternative hypotheses.

First, as mentioned above, the quantitative details of the helical trajectory are important, but these are not well-matched. The authors say that their trajectories match the experimental helical diameter and period of the trajectories but not the pitch. The results are strongly dependent on the angle between the bundles, which is used as a fitting parameter. They say that they believe that the pitch could be matched eventually by fitting the opening angle and flagella length but do not actually do it.

Second, changing directions of bundles is also likely able to produce many different types of helical trajectories with varying pitch and radii. These have not been eliminated as possibilities.

Third, the fast reorientation in the simulations is interesting, but again, it is not ruled out whether other configurations could also yield similarly fast reorientations.

Fourth, the speed of the swimming is used as supporting evidence, but this is achieved by increasing the torque on the motor (to a value somewhat higher than some report for *E. coli*, but not out of the realm of possibility). If one is allowed to adjust the motor torque, then any speed can be reached for any configuration.

Reviewer #2:

The manuscript reports results of a detailed experimental and numerical study of the swimming motion of magnetotactic cocci, bilophotrichous bacteria with flagellar bundles at both poles. The experiments show that bacteria swim along a double helical path, with a very high swim speed and short reorient time compared to other bacteria. Hydrodynamic modelling demonstrates that this is due to a pushing and a pulling bundle. Experimental and numerical results are in good semi-quantitative agreement.

I think this is a very nice investigation, which carefully elucidates the complex swimming motion of a bilophotrichous bacterium. Experiment and hydrodynamic simulation complement each other very well to characterize the geometry of the flagellar organization and swim pattern. Thus, I strongly support publication of this manuscript in *eLife*.

I have only a few comments, which the authors should consider before publication:

1) In the main text, the authors talk about flagella as well as flagella bundles. I found this pretty confusing, until I saw Figure S1. I recommend clarifying this point early in the main text.

2) The reorientation angle could be discussed in a bit more detail. As far as I understand, as one bundle changes its direction of rotation, both bundles are in pushing (or pulling) mode, which leads to a (roughly) 90° reorientation of the swimming direction. This would explain the particular value of the reorientation angle. This seems to be somewhat similar to the behavior seen in simulations of the early stages of swimming and bundle formation of peritrichous bacteria, when the flagella are initially pointing in arbitrary directions, see, J. Hu et al., Sci. Rep. 5, 9586 (2015). However, the bundle rotation direction has to change back. This would result in another 90° angle, but uncorrelated with the first. Please clarify.

Reviewer #3:

The authors present a very thorough study of the exceptional swimming capacities in terms of speed and possibilities to suddenly change direction, displayed by a magnetotactic bacterium strain (MC-1). They use a holographic method to track in 3D the swimming trajectories and fast camera observations of the rotating body to visualize the positions of the flagellar bundles hooked to the cell. They discover the presence of a double helical path characterizing the swimming kinematics that can be explained by two sets of pushing and pulling flagella bundles working cooperatively and positioned within the same hemisphere of the spherical body. They simulate a hydrodynamic model to corroborate their finding, yielding very reasonable quantitative agreement with the experiments. The model, when set to some limits also helped to discover a new swimming pattern under the application of a magnetic field. Importantly, they also find that to reach such performances, the flagella must assume motive torque and bundle rigidity significantly larger than what is usually obtained for other bacterial strains.

I found the experimental study excellent with important conclusions on an original propulsion model. My opinion is that the paper could deserve publication in *eLife*. My only concern is sometimes on the pedagogy and clarity of the explanations delivered. I believe this could be significantly be improved. I do not deny that the authors really tried to make an effort to deliver their message, in particular visually. However, it remains that sometimes, it took me several subsequent readings of the same paragraph to really understand what they actually meant. Figure 3 is a particular example of that, and I found in this instance, very hard to follow the reasoning in the caption in association with the visual message. Then I would suggest that the paper could significant gain in pedagogy and impact, if it was critically read by someone who could help to clarify some explanatory sentences.

---

## [Author Response]

Reviewer #1:[…]1) In the dark field imaging, bright spots on the cell body are identified with flagella. This identification is not obvious to me, and it is quite puzzling why the authors do not actually image the flagellar bundles. If the bundles were indeed extended from the body in front and behind the cell, I would expect that they could be visualized which would definitively prove their proposed configuration. In Son, Guasto and Stocker, 2013 sheathed flagella are quite readily imaged. In my experience, there may be difficulty seeing flagella with darkfield at 1640 fps, but they should be readily imaged at <200 fps using their camera. Note that 200 fps is more than fast enough to resolve the larger helical motion with period 72 ms. It is unclear if the authors attempted this. If the authors tried and were not able to see a leading and lagging bundle, I would take that as evidence against their claim.

We agree on the fact that a clear image of the flagella would be the easiest way to proof our claims. We agree that we should clarify that we were not able to directly observe the full flagella bundles in motion. This, however, has proven to be challenging even for larger cells that additionally move one order of magnitude slower than the here analyzed species (e.g. low signal-to-noise ratio in Son, Guasto and Stocker, 2013).

We wrote:

“The flagella bundle morphology and movements were imaged in transmission electron microscopy (TEM) and in high-intensity dark-field video microscopy (Figure 3, Video 1 and Video 2). […] This is the result of the combination of the small size of the cells and their extraordinary high swimming speed and possibly their strong flagella bundle movement. “(Result section, fourth paragraph)

To explain the flagella images in an understandable way, we introduced a new Figure 3, focusing on convincingly identifying the tracked spots on the cell surface as parts of flagella bundles.

We thereafter wrote:

“Despite of the difficulties in imaging the flagella bundles in full length, the position of the flagella bundle near the cell surface were tracked together with the cell’s trajectory over 85 ms, which corresponds to 1.6 periods on the large helical trajectory of the cell (Figure 4). […] Additionally, the bright spots’ movement pattern featured the same periodicity as the large helical swimming track of the cell (also compare Figure 3C and Figure 3D).” (Result section, fifth paragraph)

And later:

“We turned to numerical simulations of the cell’s swimming behavior, to develop a deeper understanding of the mechanisms of propulsion and rapid reorientation and to compensate the missing information gathered from flagella bundle imaging.” (Result section, sixth paragraph)

2) The authors describe the double helical path as "unexplored" in the Introduction, but based on my knowledge, double helical paths are what should be expected from most types of propulsion by flagellar bundles, with the main distinguishing feature being the size of the larger helix. The smaller helix has in the past been attributed to the rotation of the bundle or flagellar helix, see Keller and Rubinow, 1976. The larger helix arises anytime there is non-axisymmetric propulsion, see Hyon, Powers, Stocker, Fu 2012 and has been observed in many species, albeit with varying sizes of the larger helix. As discussed in point 3 below, an important implication of this is that the bar for numerical simulations cannot be simply qualitative as in "straight" vs. "double-helical," but must involve some quantitative analysis of the trajectory pitch and radius.

Thank you for the clarification. We rewrote the corresponding section (not describing double helical paths as “unexplored”).

We edited the sentence:

“Here, we confirm that MC-1 cells reach speeds of over 500 μm s^-1^ (Figure 1D) and observe that the cells travel along a double helical path, which has not been reported for bilophotrichous cells so far.” (Introduction, final paragraph)

Towards the quantitative analysis of trajectory and pitch: We have greatly extended the subsection “Discussion of pitch differences between simulation and experimental observation”.

3) The numerics have not been described in a way that clearly rules out alternative hypotheses.

As reviewer 1 correctly pointed out, we should discuss all possible varieties of different asymmetries in flagella bundle configuration in the paper. Five possible configurations are: A difference in flagella length, a difference in motor strength, an asymmetry in the equilibrium angle of the two flagella relative to the cell surface, a time-dependent movement of the flagella equilibrium angles relative to the cell surface and the initially proposed asymmetry in sense of rotation (‘pusher-puller’ hypothesis). Finally, the best match between experiments and simulations has been found for our initially claimed ‘pusher-puller’ hypothesis.

We changed the introduction to the simulation paragraph:

“A large helical path of a microswimmer is produced from an off-axis (relative to the swimming direction) torque which continuously changes the direction of the thrust force. […] Although not fully reaching experimentally observed helix pitches, the asymmetry in sense of motor rotation was the only scenario producing significant matches in helix diameter and cell speeds.” (Result section, sixth paragraph)

Further, we added the mentioned scenarios to the text:

“1-Asymmetry in senses of rotation of motors: This asymmetry is described in the original version of the submitted manuscript. […] So, the results show that asymmetry in flagella direction cannot generate a helix comparable to MC-1’s large helix.” (Subsection “Test of different asymmetries in flagella configurations”)

First, as mentioned above, the quantitative details of the helical trajectory are important, but these are not well-matched. The authors say that their trajectories match the experimental helical diameter and period of the trajectories but not the pitch. The results are strongly dependent on the angle between the bundles, which is used as a fitting parameter. They say that they believe that the pitch could be matched eventually by fitting the opening angle and flagella length but do not actually do it.

We further investigated the parameter space within the pusher-puller scenario and extended the section. More precisely, we varied the opening angle, the flagella length and the motor torque and list the resulting track parameters in the subsection “Discussion of pitch differences between simulation and experimental observation”:

*“*While the pitch of the simulation result, presented in the main text, is off by a factor of 2, the diameter and period time of helical trajectories are in good agreement with the experiments. […] It can be concluded that the motor torque can be one of the parameters that may help matching between experiment and simulation, although the definitive match was not reached in this study.”

Second, changing directions of bundles is also likely able to produce many different types of helical trajectories with varying pitch and radii. These have not been eliminated as possibilities.

In the previous simulations, we fixed the opening angle for each simulation but the direction of the flagella axes are free to be determined by the dynamics of the system. Therefore, we did not set this parameter manually and the physical model implies the direction of flagella axes based on the physical conditions such as hydrodynamic interaction and other forces in the system. According to the reviewer’s suggestion, we also did additional simulations where the direction of flagella was changed. We observe a very weak second helix in the swimming trajectories with diameter of about an order of magnitude smaller than the experimental values. The weak effect of flagella direction may originate from the flagella’s flexibility. Due to flagella’s flexibility, the two flagella approach each other at their end and the swimming of the bacteria is not very different from the symmetric case. As mentioned above, it is the dynamic of the system that determines the shape and orientation of the flagella. So, the results show that asymmetry in flagella direction cannot generate a helix comparable to the MC-1 large helix.

We described this point in the new version of the manuscript, as described in the new paragraph on different flagella bundle scenarios, as already presented.

Third, the fast reorientation in the simulations is interesting, but again, it is not ruled out whether other configurations could also yield similarly fast reorientations.

We edited the main text and added the section “Transient flagella configurations that produce fast reorientation events”. We found that the originally suggested scenario produces best fits between experiments and simulations.

We wrote in the main text:

“Our model suggests a pushing flagella bundle together with a pulling flagella bundle rotating in opposite senses. […] A transient bending-like deformation was observed in the simulated flagella bundles during an event, which made the fast reorientations possible.” (Results section, ninth paragraph)

We wrote in the text:

“Three possible transient flagella configurations for reorientation events were tested in the simulations (CCW meaning counter-clockwise and CW meaning clockwise).

[…] Based on these results, we concluded that the reorientation angle statics produced by CCW&CW → CCW&CCW → CCW&CW matched the best to the experiment result.” (Subsection “Transient flagella configurations that produce fast reorientation events”)

Fourth, the speed of the swimming is used as supporting evidence, but this is achieved by increasing the torque on the motor (to a value somewhat higher than some report for E coli, but not out of the realm of possibility). If one is allowed to adjust the motor torque, then any speed can be reached for any configuration.

Each MC-1 sheathed bundle of flagella contains several flagella. Since we modeled a sheathed bundle of flagella by a single flagellum, we considered its effective torque to be several times larger than a single motor of *E. coli*. Unfortunately, there is no reported experimental value for the motor torque of MC-1. Therefore, since we have a bundle of flagella, we tested high torques and tune it in a way that the simulated velocity fit to the experimental values.

We described more clearly in the manuscript the above reasons for why we used higher motor torques for MC-1 than a single motor of *E. coli*. We wrote:

“The parameters for bending rigidity, torsion rigidity and torque of the flagella bundles were adjusted to reproduce the observed movement characteristics (number of small helices per large helix and velocities). […] Only this assumption allowed for stable and high swimming velocities in the simulations, indicating that the function of the flagella bundle is to combine high torques with high rigidities.” (Results section, seventh paragraph)

Reviewer #2:[…]1) In the main text, the authors talk about flagella as well as flagella bundles. I found this pretty confusing, until I saw Figure S1. I recommend clarifying this point early in the main text.

This was indeed confusing. We changed all “flagella” for “flagella bundle” and moved Figure S1 to Figure 3A. Thank you for the hint.

2) The reorientation angle could be discussed in a bit more detail. As far as I understand, as one bundle changes its direction of rotation, both bundles are in pushing (or pulling) mode, which leads to a (roughly) 90° reorientation of the swimming direction. This would explain the particular value of the reorientation angle. This seems to be somewhat similar to the behavior seen in simulations of the early stages of swimming and bundle formation of peritrichous bacteria, when the flagella are initially pointing in arbitrary directions, see, J. Hu et al., Sci. Rep. 5, 9586 (2015). However, the bundle rotation direction has to change back. This would result in another 90° angle, but uncorrelated with the first. Please clarify.

The observations showed reorientation times shorter than 5 ms. In simulation, we showed that one MC-1 reorientation event can be produced by a three-step deformation of flagella (pusher&puller → pusher&pusher → pusher&puller). The reorientation angle is measured by finding the angle between the first and third steps (pusher&puller steps) irrespective of the trajectory of the middle transient pusher&pusher step. The reviewer’s reasoning for having two uncorrelated 90° reorientation is true if we give the middle step enough time such that the flagellum can find a stable configuration but since this middle step is very short, it is not possible to measure a definite angle between the first and second or the second and third steps.

We clarified the definition of reorientation angle and how we calculated it in the manuscript. We edited the following paragraph:

“Our model suggests a pushing flagella bundle together with a pulling flagella bundle rotating in opposite senses. […] A transient bending-like deformation was observed in the simulated flagella bundles during an event, which caused the fast reorientations.” (Results section, ninth paragraph)

Reviewer #3:[…]I found the experimental study excellent with important conclusions on an original propulsion model. My opinion is that the paper could deserve publication in eLife. My only concern is sometimes on the pedagogy and clarity of the explanations delivered. I believe this could be significantly be improved. I do not deny that the authors really tried to make an effort to deliver their message, in particular visually. However, it remains that sometimes, it took me several subsequent readings of the same paragraph to really understand what they actually meant. Figure 3 is a particular example of that, and I found in this instance, very hard to follow the reasoning in the caption in association with the visual message. Then I would suggest that the paper could significant gain in pedagogy and impact, if it was critically read by someone who could help to clarify some explanatory sentences.

We corrected the English in many cases and clarified the flagella imaging part. Since Figure 3 was difficult to follow, we added a new figure (now Figure 3), as discussed in the answers to reviewer 1.

We also changed the legend of the former Figure 3 (now Figure 4) to:

“Figure 4. Positions of flagella bundles on the cell over 85 ms at 1424 fps. (A) The cell swam from top to bottom in the field of view. Moving bright spots were identified as the parts of the flagella bundles that were closes to the cell surface. The cell’s helical traveling path had a period time of 52 ms. (B) The positions were tracked over time and are depicted relative to the center of the cell, represented by different spheres for different time intervals. (C) The schematic of the cell’s large helical track is color-coded according to the different time intervals where the flagella bundle parts were visible to highlight the periodicity of the flagella bundle movement.”